# Improving the diagnostic recognition of thoracic endometriosis: Spotlight on a new histo-morphological indicator

Okechukwu Charles Okafor[1]*, Ndubueze Ezemba[2], Nnaemeka Thaddeus Onyishi[3], Kevin Nwabueze Ezike[4]

1 Department of Morbid Anatomy, University of Nigeria Teaching Hospital, Ituku/Ozalla, Enugu State, Nigeria, 2 Division of Cardiothoracic Surgery, National Cardiothoracic Centre, University of Nigeria Teaching Hospital, Ituku/Ozalla, Enugu State, Nigeria, 3 Histopathology Department, Enugu State University College of Medicine, Enugu, Enugu State, Nigeria, 4 Department of Anatomic Pathology and Forensic Medicine, College of Health Sciences, Nile University of Nigeria, Abuja, Nigeria

* okechukwu.okafor@gmail.com

**Data Availability Statement:** All relevant data are within the manuscript and its Supporting Information files which are available from the figshare database (DOI number(s) doi:10.6084/m9.

## Abstract

The diagnosis of thoracic endometriosis (TE) is challenging, hence resulting in under-diagnosis as well as long delays before arriving at a correct definitive diagnosis. Our aim is to review the histopathological findings in TE, summarise the diagnostic features, identify any major histo-morphological indicator(s) hitherto unrecognised as such, suggest diagnostic criteria; all with the aim of improving the diagnostic capacity and reducing observer error even where the clinical suspicion is low. A case-control study in which a search in the pathology archives of a referral hospital over a 10-year period was conducted. Twenty-six cases of TE were identified, reviewed, and compared with a control population of 48 cases taken from common benign thoracic diseases. Nine notable histological features were identified in varying permutations in the test group, namely: endometrioid glands, lymphoid clusters, ceroid macrophages, siderophages, cholesterol crystals, capillary congestion, multinucleated giant cells, smooth muscle bundles and fibrosis. The first 6 features were frequent; each being present in over 13 (13/26; 50%) test cases. The first 8 features showed significant association with TE by the Chi-squared test (*P*<0.05). In this group, the strength of association is high for the first 4 features (Cramér's V≥0.5). The presence of ceroid macrophages is shown to be a novel key feature, previously unrecognised as such, for the identification of TE. The presence of any three of four features including endometrioid glands, lymphoid clusters, ceroid macrophages and siderophages is a suggested criterion for the definitive diagnosis of TE.

## Introduction

Thoracic endometriosis (TE) is defined as the presence of endometrioid tissue within the thoracic cavity. The aetiological mechanisms of this syndrome are not well understood [1, 2]. It presents with four well-recognised clinical entities namely, catamenial pneumothorax,

figshare.14067221.v1, doi:10.6084/m9.figshare.14102849.v1).

**Funding:** The authors received no specific funding for this work.

**Competing interests:** The authors have declared that no competing interests exist.

catamenial haemothorax, catamenial haemoptysis, and lung nodules; but less frequent presentations have been reported. The diagnosis of TE is always a challenge and for this reason it is believed to be underdiagnosed. Also diagnosis is often delayed until the symptoms' temporal relationship with menses is recognised [3]. Some authors even consider it to be a diagnosis of exclusion [4, 5]. The most consistent, albeit retrospective, series on TE included 110 patients and showed that the mean age at presentation was about 35 years, with a range from 15 to 54 years [6]. The reported interval between the onset of symptoms and establishment of a definitive diagnosis of TE ranges between 8 and 16 months [4, 6, 7].

Thoracic endometriosis is generally thought to be extremely rare, but specific figures are scanty and come almost completely from case reports and small retrospective series. Is this rarity the true scenario or is it a manifestation of under-diagnosis both on the part of the clinicians and the pathologists? In order to resolve this dilemma, the finding, on one hand, of long duration prior to diagnosis in spite of hospital presentation (possibly to different clinicians) will point to the clinicians as the cause of under-diagnosis. On the other hand, the correct and definitive diagnosis coming only after clinico-pathological reviews necessitated by non-resolution of symptoms, insistence of the clinicians or requests for second opinion will point to the histopathologist as the cause of the underdiagnosis and apparent rarity of the disease. The scenario is further compounded by a finding that less than 50% of laparoscopic biopsies of lesions visually diagnosed as endometriosis were confirmed to be endometriosis on histological study [8]. Literature suggests that both factors are involved and several studies have attempted to investigate this diagnostic entity [6, 9–12].

The histopathological features of endometriosis are the presence of endometrioid glands and stroma [13]. However, in contrast to the more common pelvic endometriosis, in TE the endometrioid glands are not always present in biopsy material. In problematic cases immunohistochemical analysis using oestrogen, progesterone, and CD10 may confirm the endometrioid glands and stroma [9, 14]. This approach holds good in research situations and when there is a high clinical suspicion of TE following suggestive symptoms; but the challenge remains as to how, in the absence of this clinical suspicion, the pathologist will be able to raise the possibility of and arrive at a definite diagnosis of TE in thoracic biopsies from women without long delays and several reviews and opinions being sought on the biopsies.

We undertook a thorough histopathological study of confirmed cases of TE, identified the constituent histo-morphological features and assessed the occurrence of these features in a control group of benign thoracic biopsy aiming to delineate novel histo-morphological indicator(s) for TE and suggest diagnostic criteria, which will improve diagnostic recognition of TE even in the absence of clinical suspicion.

## Materials and methods

This is a descriptive case-control study of thoracic biopsies seen at the University of Nigeria Teaching Hospital, Ituku-Ozalla, during a 10-year period between January 1, 2011 and August 31, 2020. The study was approved by the Health Research Ethics Committee of the University of Nigeria Teaching Hospital, Ituku-Ozalla (certificate number: NHREC/05/01/200BB-FWA00002458-1RB00002323). A search of the departmental archives was undertaken and the cases, consisting of biopsies previously diagnosed and confirmed as TE, were identified. Data, including gender, age, and clinical presentation were extracted from the histopathology reports. All identifying information in the data, including names, initials, and hospital numbers, were removed before access and inclusion into this study. A meticulous histomorphological analysis of the corresponding H& E slides of the cases was done and this yielded a complete histological portrait of the TE cases (S1 Table). Following this, a control group of

diverse thoracic biopsies, consisting of biopsies from the thoracic wall, diaphragm, lung and mediastinum was similarly selected and was assessed for the presence of the same histomorphological features earlier identified in the TE group.

The Chi-squared test of independence and Cramér's V test were done to determine association and strength of association respectively between TE, and the identified histological features when compared with other thoracic biopsies. P-value of 0.05 or less was considered statistically significant. Cramér's V coefficient was classified as high association ($V \geq 0.5$), medium association ($0.5 > V \geq 0.3$) or low association ($0.3 > V \geq 0.2$).

For each test or control case, a component score of 1 or 0 was given accordingly for the presence or absence each histological feature (S2 Table). Similarly, for each case three different group scores were calculated: the first from the sum of the scores of all the histological features assessed, the second from sum of the scores of those histological features that show association to TE based on the Chi-squared statistics, and the third from the sum of the scores of those histological features that have a high association based on Cramér's V coefficient (columns GpOmnibus, GpB and GpC respectively of S2 Table).

The area under curve of the receiver operating characteristics curve (AUC-ROC) was calculated for each of the 3 group scores in order to compare their overall usefulness in the diagnosis of TE; 1 being the best possible AUC-ROC statistic and 0.5 being the worst possible for any diagnostic criterion.

The data were analysed using R statistical software including ggplot2, pROC, vcd, and DT packages [15–19]. Results were presented in tables and charts.

## Results

Twenty-six histologically confirmed cases of TE were included in the test group. The women aged between 20 and 50 years (mean = 32.7; standard deviation [SD] = 6.6). Twenty-five of the lesions were located in the pleura, 1 with lung involvement and 1 in the pericardium alone. The mean interval between the onset of symptoms and definitive histopathological diagnosis was 49 weeks (range = 1 week to 300 weeks). Forty-eight negative controls were obtained from the same time interval of consideration from other thoracic specimens in 40 males and 8 females, aged between 2 and 75 years (mean = 40.4; SD = 17.7), with a diagnosis other than endometriosis as shown in Table 1. Only non-neoplastic lesions were included as control cases for the study.

The thoracic presenting symptoms in the test subjects are summarised in Table 2. Furthermore, some of the test patients also presented with extra-thoracic signs and symptoms like massive ascites, cyclical abdominal swelling/fullness, umbilical mass, infertility and weight loss. The treatment applied to the test population was multimodal and included hormonal therapy, tube thoracostomy, pleurodesis using 2g of tetracycline, repair surgery and pleurectomy. The treatment combination given was based on the presenting symptoms and patient response to initial therapy. The patients were followed up for a period ranging from 6 months to 5 years in both the thoracic and gynaecological outpatient clinics. Over this period only 3 patients had recurrent symptoms of TE beyond 6 months of initial intervention and all were symptom-free by 4 years. Some patients with extra-thoracic signs and symptoms like massive ascites and infertility continued to have recurrence of these extra-thoracic features beyond the follow-up period.

Grossly, cystic lesions measuring between 2mm and 10mm were seen in 4 confirmed TE specimens; the rest of the test cases had nondescript gross appearances. Nine notable histological features were identified in varying proportions in the test (TE) population, namely endometrioid glands, lymphoid clusters, ceroid macrophages, siderophages (haemosiderin-laden

**Table 1. Demographic information on all the subjects used in the study.**

| Disease entity | Status | Count | Male | Female | Age range (years) |
|---|---|---|---|---|---|
| Endometriosis | Test | 26 | 0 | 26 | 20–50 |
| Tuberculous pericarditis | Control | 4 | 2 | 2 | 24–72 |
| Nonspecific pericarditis | Control | 16 | 14 | 2 | 16–62 |
| Tuberculous pleuritis | Control | 6 | 6 | 0 | 25–52 |
| Nonspecific pleuritis | Control | 7 | 4 | 3 | 55–63 |
| Pulmonary tuberculosis | Control | 1 | 0 | 1 | 58 |
| Empyema thoracis | Control | 9 | 9 | 0 | 2–67 |
| Histoplasmosis (pleural and pulmonary) | Control | 1 | 1 | 0 | 50 |
| Hypersensitivity pneumonitis | Control | 1 | 1 | 0 | 75 |
| Necrotizing pneumonia | Control | 1 | 1 | 0 | 2 |
| Organizing haematoma (pleural and pulmonary) | Control | 1 | 1 | 0 | 35 |
| Aspergillosis (pleural and pulmonary) | Control | 1 | 1 | 0 | 36 |
| **TOTALS** | | **74** | **40** | **34** | |

macrophages), cholesterol crystals, smooth muscle bundles, capillary congestion, multinucleated giant cells and fibrosis (S1 Table; Fig 1A–1I).

Twenty-four cases (24/26; 92%) showed endometrioid glands but this was sometimes very focal and required the examination of several sections before they could be identified (Fig 2). The glands were either inactive or slightly proliferative; no secretory gland was seen. Occasionally, they appeared atrophic and were lined by a single layer of cuboidal or attenuated epithelial cells. Hobnail and clear cells were the metaplastic changes frequently seen. Two cases (2/26; 8%) (Lines 13 and 17 of S1 Table) lacking endometrioid glands were classified as stromal endometriosis with siderophages, ceroid macrophages, cholesterol crystals, lymphoid clusters, capillary congestion and fibrosis. In addition, these 2 patients had presented with catamenial chesty symptoms suggestive of TE. Bundles of smooth muscle, distinct from the wall of blood vessels, were seen in 4 cases (4/26; 15%).

Three histological features, namely lymphoid clusters, fibrosis and capillary congestion, were present in all the 26 test cases (26/26; 100%) (Fig 2). Overall, there are 6 frequent features, adding siderophages, endometrioid glands and ceroid macrophages to the earlier 3, which, individually are present in the majority of the cases (>13/26; 50%); all 6 together are present in 16 cases (16/26; 62%); and in different permutations are present in all 26 cases (26/26; 100%). These 6 frequent features will be referred to as Group A.

The percentage frequency distribution of all 9 histological features in the test and control populations were respectively compared and shown in Table 3. Not being exclusively present

**Table 2. Summary of the clinical signs and symptoms in the test population.**

| Thoracic signs and symptoms | n/N (%) |
|---|---|
| Massive haemothorax | 18/26 (69.2) |
| Catamenial chest pain/cough | 10/26 (38.5) |
| Catamenial pneumothorax | 6/26 (23.1) |
| Catamenial breathlessness | 3/26 (11.5) |
| Hydropneumothorax | 2/26 (7.7) |
| Catamenial haemoptysis | 1/26 (3.8) |
| Catamenial haematemesis | 1/26 (3.8) |
| Atypical chest pain | 1/26 (3.8) |

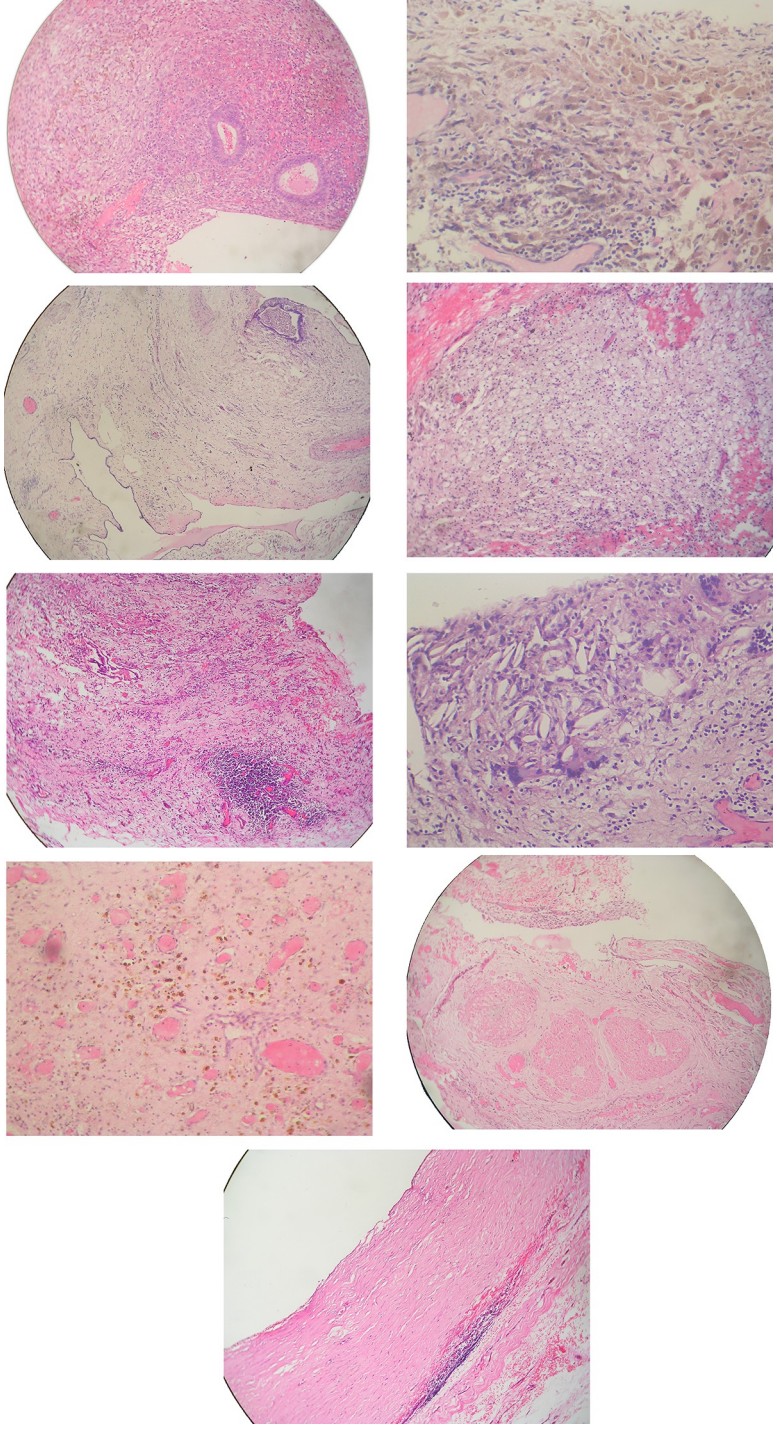

**Fig 1.** A: Typical thoracic endometriosis simulating native proliferative endometrium. H&E staining, original magnification X300. B: Dilated endometrioid gland with intraluminal pseudoxanthoma cells; the surrounding stroma is inconspicuous. H&E staining, original magnification X150. C: Lymphocytic aggregate and dilated and congested blood vessels in oedematous nonspecific stroma; a distorted endometrioid gland is also present. H&E staining, original magnification X100. D: Oedematous hyalinised stroma with congested capillaries and siderophages with golden-brown pigments. H&E staining, original magnification X300. E: Ceroid macrophages with brownish finely granular cytoplasm. H&E staining, original magnification X300. F: Foamy macrophages in the stroma. H&E staining, original magnification X150. G: Cholesterol crystals and multinucleated foreign body-type giant cells. H&E staining, original magnification X300. H: Bundles of smooth muscle cells. H&E staining, original magnification X150. I: Pleural fibrosis. H&E staining, original magnification X150.

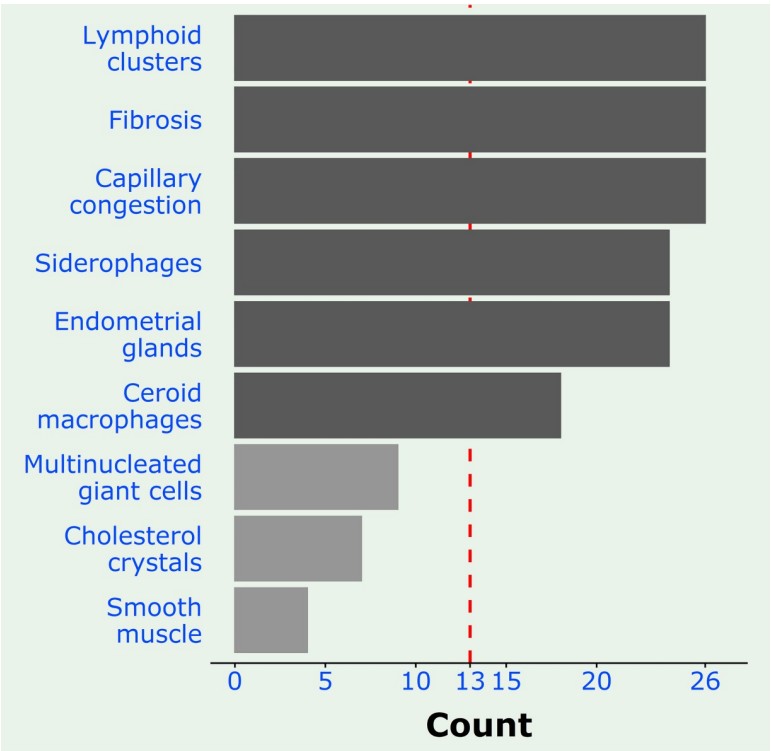

**Fig 2. Barplot showing the frequency distribution of the various histological features in the test (thoracic endometriosis) population.** The stripped red line marks the point of separation between the frequent or Group A features (darker bars) affecting more than 13 (>13/26; 50%) patients from the features (lighter bars) affecting less than 13 (<13/26; 50%) patients.

in the test population, these features were respectively analysed in both populations using Chi-squared statistical test for independence. A statistical difference ($P<0.05$) was observed in 8 histological features (endometrioid glands, lymphoid clusters, ceroid macrophages, sidero-phages, cholesterol crystals, smooth muscle bundles, capillary congestion and multinucleated giant cells) suggesting that these features could be associated with the TE. These will hence-forth be referred to as Group B. Fibrosis alone, with $P$ value of 0.088, did not show any statisti-cal difference between the test and control groups and this suggests that it is not reliably associated with the TE.

The result of the Chi-squared test of independence was further tested with the Cramér's V statistical method to measure the strength of association in those 8 features (Group B) that were significantly associated with the TE (Fig 3). Four features, notably endometrioid glands, lymphoid clusters, ceroid macrophages and siderophages, demonstrate a high degree of associ-ation with TE, and will henceforth be referred to as Group C. Three features, notably choles-terol crystals, smooth muscle and capillary congestion, show medium association. The presence of multinucleated giant cells has a low association with endometriosis.

To summarise the histological findings, each patient was given total scores based on these 3 groups of histological features, namely: (i) Group Omnibus: the sum of all 9 features (GpOm-nibus column of S2 Table); (ii) Group B: the sum of the 8 features with significant association to TE based on the Chi-squared statistics obtained in this study (GpB column of S2 Table); and (iii) Group C: the sum of the 4 features with high association with TE based on Cramér's V coefficient obtained in this study (GpC column of S2 Table). Fig 4 shows a boxplot of the distribution of the scores based on these groups.

**Table 3. Comparison of percentage frequency distribution of histological features in the test and control populations, along with the corresponding Chi-squared test statistic of independence, its *P*-value and the Cramér's V coefficient for strength of association.**

| Histological feature | Percentage | X[2a] | *P*-value[b] | V[c] |
|---|---|---|---|---|
| Endometrioid glands | | 65.5 | <0.001 | 0.94 |
| Test | 92.3 | | | |
| Control | 0.0 | | | |
| Lymphoid clusters | | 47.2 | <0.001 | 0.80 |
| Test | 100 | | | |
| Control | 16.7 | | | |
| Ceroid macrophages | | 43.9 | <0.001 | 0.77 |
| Test | 69.2 | | | |
| Control | 0.0 | | | |
| Siderophages | | 34.7 | <0.001 | 0.69 |
| Test | 92.3 | | | |
| Control | 20.8 | | | |
| Cholesterol crystals | | 14.3 | <0.001 | 0.44 |
| Test | 26.9 | | | |
| Control | 0.0 | | | |
| Smooth muscle bundles | | 7.8 | 0.005 | 0.32 |
| Test | 15.4 | | | |
| Control | 0.0 | | | |
| Capillary congestion | | 7.0 | 0.008 | 0.31 |
| Test | 100 | | | |
| Control | 77 | | | |
| MGC [d] | | 4.0 | 0.045 | 0.23 |
| Test | 34.6 | | | |
| Control | 14.6 | | | |
| Fibrosis | | 2.9 | 0.088 | 0.20 |
| Test | 100 | | | |
| Control | 89.6 | | | |

[a] Chi-squared statistic for independence.

[b] *P*-value of the Chi-squared statistic.

[c] Cramér's V coefficient for strength of association.

[d] Multinucleated giant cells.

The usefulness of the three group scores for the diagnosis of TE were found to be nearly identical (AUC-ROC of 0.995, 0.994 and 0.996 for Groups Omnibus, B, and C respectively) as well as to be very good indices for disease diagnosis [18]. It is notable that the AUC-ROC of Group C is marginally superior to that of the other 2 groups. For each group the best cutoff point was determined as the next score above the highest score of the control group i.e., that score at which there is no false positive case (specificity = 1) (S2 Table). This cutoff score was 5, 4 and 3 for Groups Omnibus, B and C respectively. At these respective scores there were 2 false negatives in each of the 3 groups, hence the sensitivity was 0.923 for all three groups.

Twenty-four of the 26 test cases had endometrioid glands and 2 did not. Both of these latter 2 presented with the other 3 Group C features namely ceroid macrophages, siderophages, and lymphoid clusters. No control case contained these 3 features together, thus confirming that even in the absence of endometrioid glands, the other 3 Group C features together could be used to diagnose TE. A similar statement can also be made of the Group B features where,

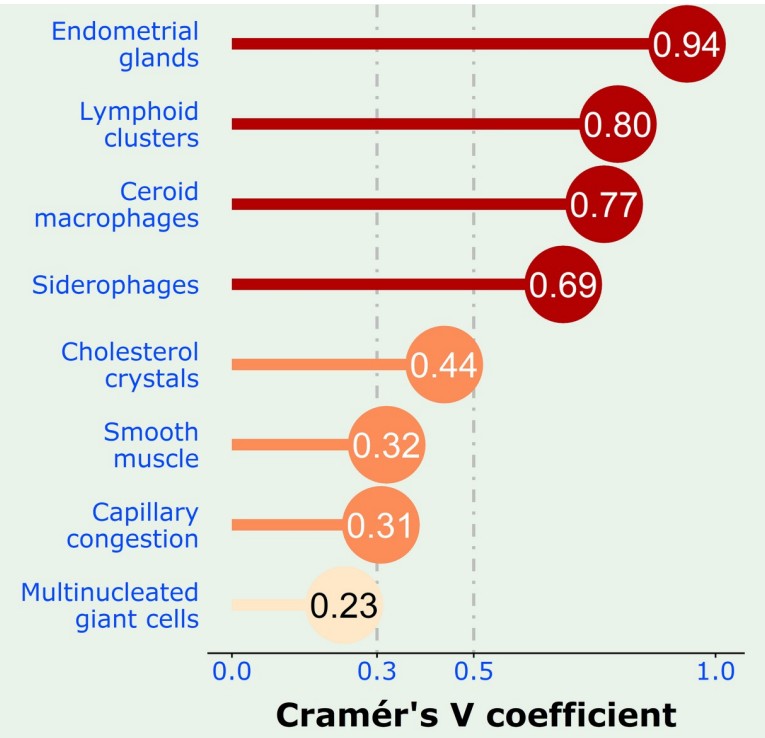

**Fig 3. Lollipop plot showing the strength of association with disease of the various histological features using the Cramér's V coefficient.** The dashed lines divide the histological features into those of high association (brick red), medium association (coral) and low association (blanched almond).

even in the absence of endometrioid glands, any 4 of the other features could be diagnostic of TE and no control case had 4 of these Group B features together.

## Discussion

The histological diagnosis of endometriosis is typically based on the presence of endometrioid glands with or without endometrioid stroma [14] (Fig 1A–1C). The glands show typical endometrioid features, usually of proliferative or inactive type. Sometimes they are cystically dilated and are grossly visible as vesicles (Fig 1B). The appearance of the endometriotic tissue varies with the duration of the process as well as relative to its response to the normal hormonal fluctuations of the menstrual cycle. It is only in 44% to 80% of cases that the corresponding cyclical changes are seen in the endometriotic tissue when compared to native endometrium in reproductive women [20, 21]. The lining cells have various appearances including columnar, cuboidal, flattened and even mildly dysplastic cells with hyperchromatic nuclei. All these changes further increase the difficulty in identification of endometriosis as such.

Small congested arterioles and capillaries (referred to as capillary congestion) are present, and provide a helpful initial clue to the identification of the endometriotic nature of the lesion. For diagnostic purposes it is important to insist that these vessels be numerous, so much so that they draw attention to the lesion at scanning magnification (Fig 1C and 1D).

With menstruation, haemorrhage may occur within the stroma and glandular lumina of endometriotic foci, as well as a secondary inflammatory response consisting predominantly of a diffuse infiltration of pigmented macrophages. These macrophages fragment and engulf the breakdown products, and can be classified into ceroid macrophages (or ceroid-laden

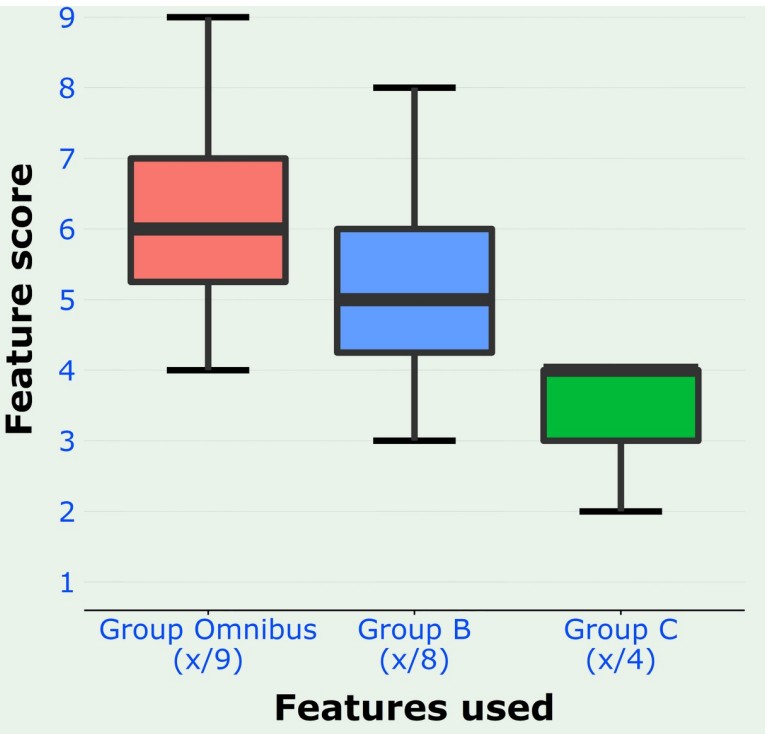

**Fig 4. Boxplot of distribution of scores based on 3 groups of features: all features (Group Omnibus), those with association to disease by Chi-squared test (Group B) and those with high strength of association by Cramér's V coefficient (Group C).** The tips of the whiskers in these boxplots represent the minimum and maximum scores of the various groups.

macrophages) and siderophages (haemosiderin-laden macrophages) based on their cytoplasmic phagocytic contents. Ceroid and related foamy lipids are lipochrome pigments and their outstanding feature is their pale foamy to eosinophilic to greyish-brown colour, sudanophilia and fine granularity (Fig 1E and 1F). Ceroid macrophages with pale foamy cytoplasm are also referred to as foamy macrophages. Haemosiderin is a form of storage iron derived chiefly from the breakdown of erythrocytes. Haemosiderin appears as a coarse, golden brown, refractile pigment with haematoxylin-eosin stain and as blue granules with Perls' Prussian blue. Some of these macrophages fuse to form multinucleated giant cells of the foreign body type (Fig 1G).

Variable numbers of lymphocytes and smaller numbers of other inflammatory cells are usually present; the former are seen as lymphoid clusters and constitute an important clue to diagnosis in difficult cases (Fig 1C). On rare occasions some of these lymphoid clusters were seen to have a germinal centre, thus confirming their reactive nature. Typically, endometriosis is associated with reactive fibrosis and, if prominent and of significant duration, can result in conspicuous adhesions (Fig 1I).

With chronicity and progression of the lesion this classical picture is further modified by necrotic and metaplastic processes, thus producing varied components such as cholesterol crystals (Fig 1G) and smooth muscle bundles [22] (Fig 1H). These features, on entering the mix, serve to render the diagnosis of endometriosis even more difficult and enigmatic, especially in small biopsies.

In some cases stroma only is found in biopsies despite exhaustive sampling of the specimens, and the preferred terminology is stromal endometriosis. This rare variant of endometriosis has also been described in the peritoneum, cervix, ovary and omentum [22, 23–25].

A high index of suspicion by the clinician as well as the experience of the pathologist are necessary for a correct diagnosis to avoid such erroneous classifications as nonspecific pleuritis, fibrinous pleuritis and similar [26]. In our series, three cases had to be re-examined by the pathologist at the request of the surgeons after the initial diagnosis of nonspecific pleuritis. In the cases of complete diaphragmatic rupture with no obvious evidence of endometriotic tissue as well as in those with multiple diaphragmatic fenestrations, it is advisable to biopsy the edges of the diaphragm in the former, and do *en-bloc* resection of the fenestrations in the latter. The defect can always be repaired using GORE-TEX patch. This approach ensures tension-free repair/reconstruction of the diaphragm in addition to providing tissue for histological examination. Similarly, when the diaphragm is intact with no endometriotic nodule, parietal pleurectomy as done in the majority of our patients, even when there are no obvious macroscopic deposits, will yield tissue for histological examination.

This particular study is 'the other side of the coin' and complementary to other studies that have attempted to correlate histological diagnosis of endometriosis with grossly visualised lesions. In earlier studies, the proportion of grossly visible lesions confirmed by histology ranged between 3.1% and 100% [27–31]. Unfortunately, differing and limited patient populations, small size of specimens, and inconsistencies in biopsy techniques have marred all of these studies. However, a detailed and prospective correlational study carried out on 44 patients with peritoneal biopsies found a positive predictive value (PPV) (percentage of positive histological findings of endometriosis among the positive visual [endoscopic] findings) of 45% and negative predictive value (percentage of normal histological findings among the negative visual findings) of 99% [8]. This study went ahead to conclude that a diagnosis of endometriosis should be established only after histological confirmation. The PPVs varied considerably among their 9 biopsy sites: highest in the cul-de-sac (65%) and lowest in the right psoas muscle (14%). It considered even the highest PPV still insufficient to justify visual diagnosis alone. Our study builds on this conclusion with the intent of improving the diagnostic capacity of the histopathological evaluation.

For bench diagnosis of TE we propose 2 new criteria as alternatives, as shown in Table 4, especially when the hitherto traditional criterion (of finding endometrial glands) is not applicable. The first, and recommended, alternative criterion, based on Group C of our study, consists of the presence of ceroid macrophages and siderophages and lymphoid clusters (total of 3 features). In addition, the identification of any one of the Group C features in a thoracic biopsy should serve as an alarm bell to the pathologist so as to further consider the possibility of TE in that case. The second and less favoured proposed criterion, based on the Group B features, consists of the presence of ceroid macrophages plus any other 3 of the other 6 histological features of Group B (total of 4 features), excluding endometrioid glands which already is a stand-

**Table 4. The traditional criterion and two new suggested alternative criteria for the diagnosis of thoracic endometriosis.** These new alternative criteria are based on the features of Groups C and B.

| Traditional criterion (1 feature only) | First alternative criterion (3 features) (based on Group C) | Second alternative criterion (4 features) (based on Group B) |
|---|---|---|
| Endometrioid gland **only** | Ceroid macrophages **AND** | **Ceroid macrophages AND** any other 3 of the following: |
| | Siderophages **AND** | (i) Siderophages |
| | Lymphoid clusters | (ii) Lymphoid clusters |
| | | (iii) Cholesterol crystals |
| | | (iv) Capillary clusters |
| | | (v) Multinucleated giant cells |
| | | (vi) Smooth muscle bundles |

alone/traditional criterion. This second alternative criterion is inferior to the first since it includes some histological features that have only a medium strength of association with TE. It must include the presence of ceroid macrophages. We do not recommend Group Omnibus as a diagnostic criterion for two reasons. Firstly, it contains a feature (fibrosis) which we prove not show any association to TE. Secondly, it consists of too many histological features for assessment and this could prove cumbersome in bench work.

Summarising the results of this study in a Venn plot (Fig 5) we can see the overlap of Groups A (frequent features), B (Chi-squared-associated features) and C (strongly-associated

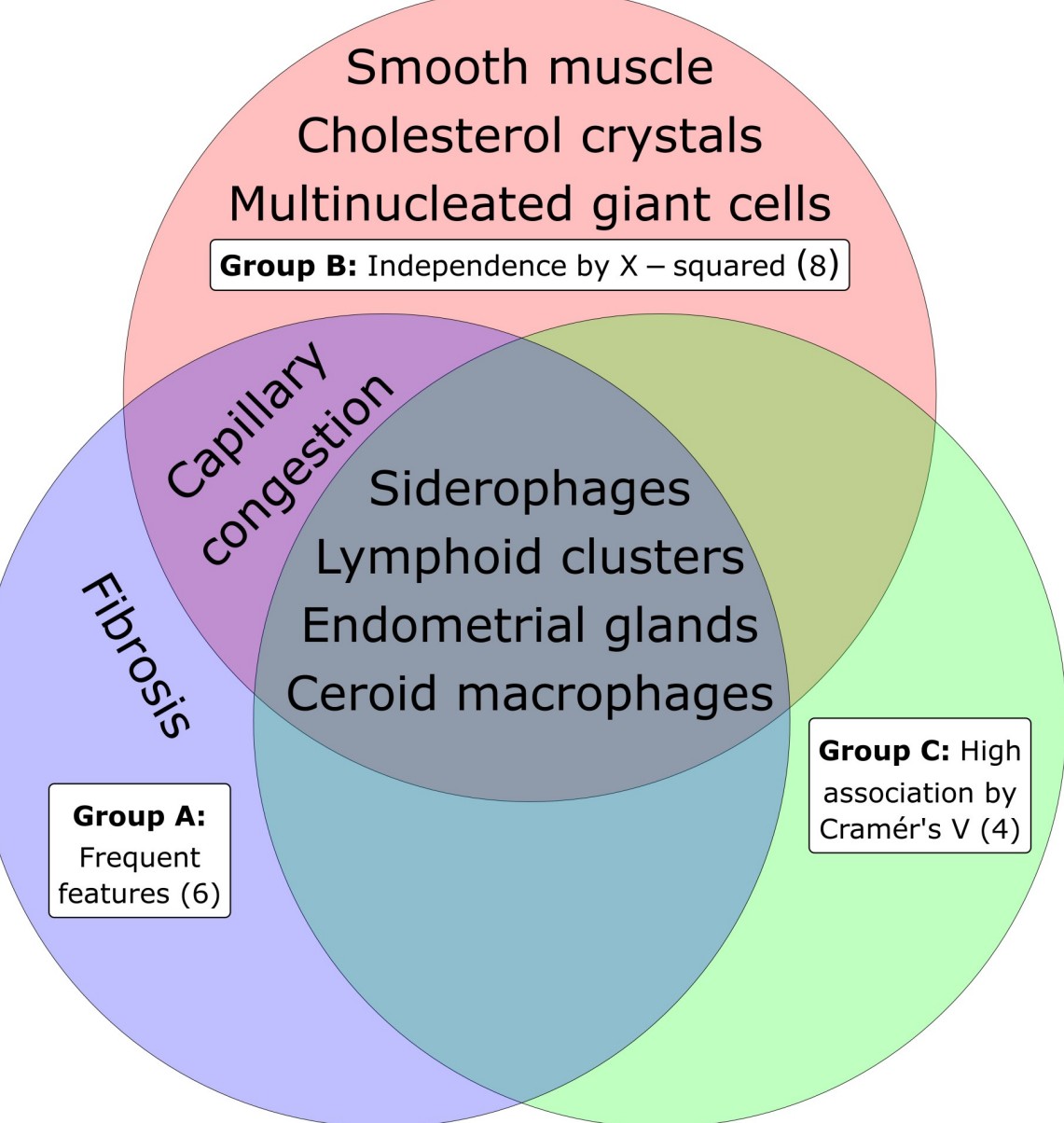

**Fig 5. Venn plot showing the overlap of the 6 frequent features (Group A), the 8 features with significant association to disease by Chi-squared test (Group B), and the 4 features with high strength of association to disease by the Cramér's V coefficient (Group C).**

by Cramér's V). It is notable that the innermost overlapping segment in the 3 circles contains all the features of Group C.

An interesting finding in this study is the emergence of the importance of macrophages, especially ceroid macrophages, in the diagnosis of endometriosis, something that had not come to light before now. Articles published recently have thrown the spotlight on the key pathogenetical role of macrophages for establishment of successful and viable colonies by endometrial fragments that accidentally reach the peritoneal cavity and other foreign sites during the fertile years [32, 33]. Macrophages are known to sense cues (e.g. hypoxia, cell death, foreign entities, etc.) in body lesions and react by delivering signals to restore the local homeostasis. Endometriosis may be due to a misperception by macrophages about ectopic endometrial tissue, which they perceive as a wound and subsequently activate programmes leading to ectopic cell survival. This has radical implications for the development of novel medical treatments of endometriosis. That said, this study opens yet another vista and suggests a key diagnostic role for these macrophages in endometriosis. The ceroid macrophages and siderophages were frequent in this study (Group A) and showed high statistical association with TE (Group C, Fig 1D–1F). One or both types of macrophages were present in 24 cases (92%). It is notable, however, that the presence of multinucleated giant cells showed only a low association with the disease and therefore does not feature among the favoured diagnostic criteria.

Traditionally the presence of siderophages has been emphasised in endometriosis of all sites; ceroid macrophages have received no special recognition [34]. Without doubt, in endometriosis, the cyclical changes of cell growth, haemorrhage and necrosis seem likely to provide generous amounts of suitable substrates for the formation of ceroid. Why then are ceroid macrophages not more commonly recognised? One possible contributing reason is that observers do not take the pains of differentiating between the refractile, golden brown, coarse, Perl-staining granules of siderophages (haemosiderin) from the pale foamy to eosinophilic to greyish-brown, finely granular, sudanophilic content of ceroid macrophages [22]. As a result of this, less extensive collections of ceroid macrophages in endometriosis have been overlooked in bench practice and under-reported in literature. Contrary to the general trend, this study shows that, compared with siderophages, ceroid macrophages show a higher degree of association with TE by the Cramér's V statistical analysis of association (0.77 versus 0.69); no control case showed any ceroid macrophages while 10 out of 48 (21%) control cases showed siderophages. We therefore emphasise ceroid macrophages over siderophages in the diagnosis of endometriosis.

The importance of having a high index of suspicion for TE in women of reproductive age who have experienced catamenial symptoms and possible recurrence of these symptoms after interventions, and also the requirement of a multidisciplinary (pulmonologist, thoracic surgeon, pathologist and radiologist) approach to diagnosis has been advocated by most authors on the subject [35, 36]. Without denying this aforementioned importance but building upon it, the findings of this study should serve as an important extra tool in the armamentarium of the pathologist in order to play yet a bigger role in the management of these TE patients i.e., both to raise a suspicion when there is none and to provide a definitive diagnosis, even in difficult cases. The earlier and more readily TE is diagnosed, the better the prognosis. The intent here is that TE should no longer be a diagnosis of exclusion.

We consider as potential limitations to this study the following: the non-identity of the gender of the test and control populations, the inequality of the size of the 2 populations, and the fact that the suggested diagnostic criteria were not evaluated in a different cohort. The total exclusion of neoplastic lesions in the thorax is a possible confounding factor given the varied inflammatory and reactive changes that accompany such lesions.

## Supporting information

**S1 Table. Table of distribution of histological features and clinical information of test and control individuals.** The columns for the histological features are named as follows: Gland = endometrioid glands; SideroP = siderophages; CeroidM = ceroid macrophages; Choleste = cholesterol crystals; SM = smooth muscle bundles; MGC = multinucleated giant cell histiocytes; LymphClus = lymphoid clusters; CappCong = capillary congestion; Fibrosis = fibrosis.
(XLSX)

**S2 Table. Table of distribution of scores for histological features and the grouping of scores by three different criteria.** The "yes" and "no" entries for the histological features of S1 Table have been converted into "1" and "0" respectively and columns for the grouping of scores have been added and named as follows: GpOmnibus = Group Omnibus; GpB = Group B; GpC = Group C.
(XLSX)

## Author Contributions

**Conceptualization:** Okechukwu Charles Okafor, Ndubueze Ezemba.

**Data curation:** Okechukwu Charles Okafor, Ndubueze Ezemba.

**Formal analysis:** Okechukwu Charles Okafor, Nnaemeka Thaddeus Onyishi, Kevin Nwabueze Ezike.

**Funding acquisition:** Okechukwu Charles Okafor, Ndubueze Ezemba, Nnaemeka Thaddeus Onyishi, Kevin Nwabueze Ezike.

**Investigation:** Okechukwu Charles Okafor, Ndubueze Ezemba, Nnaemeka Thaddeus Onyishi, Kevin Nwabueze Ezike.

**Methodology:** Okechukwu Charles Okafor, Ndubueze Ezemba, Nnaemeka Thaddeus Onyishi.

**Project administration:** Okechukwu Charles Okafor, Nnaemeka Thaddeus Onyishi, Kevin Nwabueze Ezike.

**Resources:** Okechukwu Charles Okafor, Nnaemeka Thaddeus Onyishi, Kevin Nwabueze Ezike.

**Software:** Okechukwu Charles Okafor.

**Supervision:** Okechukwu Charles Okafor, Ndubueze Ezemba, Kevin Nwabueze Ezike.

**Validation:** Okechukwu Charles Okafor, Kevin Nwabueze Ezike.

**Visualization:** Okechukwu Charles Okafor, Ndubueze Ezemba, Nnaemeka Thaddeus Onyishi, Kevin Nwabueze Ezike.

**Writing – original draft:** Okechukwu Charles Okafor.

**Writing – review & editing:** Okechukwu Charles Okafor, Ndubueze Ezemba, Nnaemeka Thaddeus Onyishi, Kevin Nwabueze Ezike.

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
