## [Decision Letter · Decision Letter 0]

18 Jan 2021

PONE-D-20-36704

Improving the diagnostic recognition of thoracic endometriosis: Spotlight on a new histo-morphological indicator

PLOS ONE

Dear Dr. Okafor,

Thank you for submitting your manuscript to PLOS ONE. After careful consideration, we feel that it has merit but does not fully meet PLOS ONE’s publication criteria as it currently stands. Therefore, we invite you to submit a revised version of the manuscript that addresses the points raised during the review process.

This study addressing novel histological features in thoracic endometriosis has been revised by three reviewers. Although they have recognized the interest of the paper, they have raised some concerns that need to be addressed. The Authors should consider all the issues raised, particularly those from Reviewer #3. The paper has too many figures. The Authors should follow suggestions from Reviewer #3 in this regard and consider to list figures as Fig 1, A,B,C ect. More importantly, the statistical analysis has to be better described in Materials and Methods. The Authors have to explain how they mentioned specificity and sensitivity of the various patterns without having calculated them based on ROC curves and AUC analysis. Words related to the statistical analysis need to be more precise.

We look forward to receiving your revised manuscript.

Kind regards,

Paola Viganò

Academic Editor

PLOS ONE

2. Thank you for including your ethics statement: "This study was approved by the institutional review board at the institution of study."  

3. In your ethics statement in the manuscript and in the online submission form, please ensure that you have discussed whether all data/samples were fully anonymized before you accessed them and/or whether the IRB or ethics committee waived the requirement for informed consent. If patients provided informed written consent to have data/samples from their medical records used in research, please include this information.

4. In the ethics statement in the manuscript and in the online submission form, please provide additional information about the patient records/samples used in your retrospective study, including: a) the date range (month and year) during which patients' medical records/samples were accessed.

5. In your Methods section, please provide additional information about the medical data/samples collected and the demographic details of the human subjects. Please ensure you have provided sufficient details to replicated the analyses such as a table of relevant demographic details.

6. In your Methods section, please provide a description of the outcomes assessed in your study.

7. Please ensure you have discussed any potential limitations of your study in the Discussion, including study design, sample size and/or potential confounders.

8. To comply with PLOS ONE submission guidelines, in your Methods section, please provide additional information regarding your statistical analyses. For more information on PLOS ONE's expectations for statistical reporting, please see https://journals.plos.org/plosone/s/submission-guidelines.#loc-statistical-reporting.

Reviewers' comments:

Reviewer's Responses to Questions

**Comments to the Author**

1. Is the manuscript technically sound, and do the data support the conclusions?

Reviewer #1: Yes

Reviewer #2: Yes

Reviewer #3: Partly

2. Has the statistical analysis been performed appropriately and rigorously? 

Reviewer #1: I Don't Know

Reviewer #2: Yes

Reviewer #3: I Don't Know

3. Have the authors made all data underlying the findings in their manuscript fully available?

Reviewer #1: Yes

Reviewer #2: Yes

Reviewer #3: Yes

4. Is the manuscript presented in an intelligible fashion and written in standard English?

Reviewer #1: Yes

Reviewer #2: Yes

Reviewer #3: Yes

5. Review Comments to the Author

Reviewer #1: I congratulate the Authors for this interesting paper. Histological diagnosis of thoracic endometriosis in not as common as the clinical diagnosis. I would suggest to the Authors to report in the Discussion section a brief view of the literature about the discrepancy between the clinical and histological diagnosis reported by different centers.

Reviewer #2: The authors reported an interesting study regarding the pathological features of thoracic endometriosis. The paper il well written and interesting. I have some minor points that should be reviewed:

- Introduction, line 46, 47: please, cite some possible theory regarding etiology of TE

- Introduction should be shorten

- Material and methods, from line 106 to 114: this part should be moved in the results part

- Discussion: I would suggest discussing also when TE is impossible to be proved by histology (e.g.: diaphragmatic hernias). For instance, recently it has been reported (Viti A, et al. Endometriosis Involving the Diaphragm: A Patient-Tailored Minimally Invasive Surgical Treatment. World J Surg. 2020 Apr;44(4):1099-1104.) that up to 45% of patient with diaphragmatic endometriosis had no histological diagnosis. Please discuss on this.

Reviewer #3: The article needs some work. And I would like to give you some suggestions.

First of all in endometriosis there are endometrioid glands and endometrioid stroma (not endometrial).

1) periods lines 63-68 are confusing; be concise.

2) lines 68-70

To support this, one study [8] found that less than 50% of laparoscopic biopsies suspicious for endometriosis were really endometriosis.

3) delete lines 76-83

Insert the following:

Histopathologic features of endometriosis include the presence of endometrioid glands, stroma and stroma; hemosiderin-laden macrophages are also commonly present (Flieder 14). However, in contrast to the more common pelvic endometriosis, in thoracic endometriosis the endometrial glands are not always present in biopsy material. In problematic cases immunohistochemical analysis using ER, PR, and CD10 may confirm the endometrioid glands and stroma.

4) Matherial and methods

insert from results:

The interval between the onset of symptoms 239 and definitive histopathological

diagnosis was 49 weeks (range = 1 week to 300 weeks).

5) Figures

There are too many figures.Delete figures 2, 4, and 11. Change the order of the rest:

Fig 1: Typicaltoracic endometriosis simulating eutopic proliferative endometrium.H&E staining, original magnification X300.

Fig 2 (ex Fig 10): Dilated endometrioidgland with intraluminal pseudoxanthoma cells; the surroundinginconspicousendometrioid stroma merges imperceptively into non specificedematous stroma. H&E staining, original magnification X150.

Fig 3 (ex 5): Lymphocytic aggregate and dilated and congested blood vessels in edematous non specificstroma; a distorted endometrioidgland is also present. H&E staining,original magnification X100.

Fig 4 (ex 3): Edematous hyalinized stroma with congested capillaries and

siderophages with golden-brown pigments. H&E staining, original magnification

X150 & X300.

Fig 5 (ex 6): Macrophages with brownishfinely granular cytoplasm (pseudoxantoma cells). H&E staining, original magnification X300.

Fig 6 (ex 7): Foamyhistiocytes in the stroma. H&E staining,original magnification X150.

Fig 7 (ex 8): Crystals andmultinucleated foreign body-type giant cells. H&E staining, original magnification X300.

Fig 8 (ex 9): Bundles of smooth muscle cells.H&E staining, original magnification X150.

Fig 9 (ex12): Pleural fibrosis. H&E staining, originalmagnification X150.

6) Results

Change the order:

Nine notable histological features were identified in varying proportions in the test

(TE) population, namely endometrial glandsand or stroma, lymphoid clusters, ceroid macrophages,siderophages, cholesterol crystals, smooth muscle bundles, capillary congestion,multinucleated giant cells and fibrosis (Figs 1 - 12).

Twenty-four cases (92%) showed endometrial glands but the finding was sometimes

focal and required the examination of several sections to be identified. The glands were either inactive or slightly proliferative; no secretory gland was seen. Occasionally, they appeared atrophic and were lined by a single layer of cuboidal or flat epithelial cells. Hobnail and clear cell metaplasia were the metaplastic changes occasionally seen. Two cases (8%) lacking endometrioid grands were classified as stromal endometriosis. with siderophages, ceroid macrophages, foreign body-type multinucleated macrophages, cholesterol crystals, capillary congestion and fibrosis. In addition these 2 patients complained of catamenial chesty symptoms suggestive of thoracic endometriosis. Bundles of smooth muscle, distinc from the wall of blood vessels, were seen i 4 cases (15%).

Three histological features, namely lymphoid clusters, fibrosis and capillary

congestion, were present in all the 26 test cases (100%) (Fig 13). Overall, there are

6 prevalent features, adding siderophages, endometrioidglands and or stroma and ceroid

macrophages to the earlier 3, which, individually are present in the majority of the

cases (13 or more cases or 50%); all 6 together are present in 16 cases (62%); and

in different permutations are present in all 26 cases (100%). These 6 features, being

prevalent, constitute the backbone for diagnosis of TE and will be referred to as

Group A.

The rest may be ok.

7) Discussion

The histologic diagnosis ofendometriosis is based onthe typical presence of both endometriotic glands and stroma. The glands almost always show typical endometrioid features, usually of proliferative or inactive type; the stroma may be typical with the spiral arteries, which may be engorged with erythrocytes, and provide a helpful initial clue to the identification of the endometriotic nature of the lesion. However, in many cases the stroma may be very subtle, confined to a thin or poorly defined often discontinuous periglandular zone; it can undergo smooth muscle metaplasia or replaced by histiocytes of “foamy type”. Small congested arterioles and capillaries (referred to as capillary congestion) may draw attention to the lesion at scanning magnification. In some cases stroma only is found in biopsies and the preferred terminology is stromal endometriosis (17, 18-20).

Similarly with abdominal or pelvic endometriosis, glandular and stroma breakdown may occur favoringe haemorrhage. Clusters of pigmented histiocytes become common, including ceroid-laden macrophages and haemosiderin-laden macrophages or siderophages. In older lesions fibrosis, lymphoplasmacellular infiltrate, with follicle formation, and foreign body reaction with cholesterol crystals are common (18-20).

A high index of suspicion by the clinician as well as the experience of the pathologist are necessary for a correct diagnosis to avoid [21] erroneous classification as nonspecific pleuritis, fibrinous, pleuritis and similar. In our series, three cases had to be re-examined by the pathologist at the request of the surgeons after the initial diagnosi of nonspecific pleuritis

Summarizing the results of this study in a Venn plot (Fig 16) we can see the overlap

of Groups A, B and C. It is notable that the innermost overlapping segment in the 3

circles contains all the features of Group C. Furthermore, at least 2 Group C features

are present in all TE cases. Given the high sensitivity and specificity to the diagnosis of TE, we recommend that Group C be used as criteria for diagnosis of thoracic endometriosis, as shown in the first column of Table 3, of which any 2 features are diagnostic of the disease. In addition, the identification of any of these 4 Group C features in any thoracic biopsy in a feature should serve as an alarm bell for considering the possibility of TE.

8) Finally I hope that another reviewer will check on statistical analisis.

6. PLOS authors have the option to publish the peer review history of their article (what does this mean?). If published, this will include your full peer review and any attached files.

Reviewer #1: **Yes: **Paola Ciriaco

Reviewer #2: **Yes: **Pietro Bertoglio

Reviewer #3: No

---

## [Author Response · Author response to Decision Letter 0]

26 Feb 2021

ACADEMIC EDITOR#1: “consider to list figures as Fig 1, A,B,C etc”

OUR REPLY: We have reduced the number of figures and modified the numbering to match the above format.

ACADEMIC EDITOR#2: “the statistical analysis has to be better described in Materials and Methods”

OUR REPLY: The statistical methods used were briefly described in Materials and Methods and the purpose for applying them stated. Lines 101-138.

ACADEMIC EDITOR#3: “explain how they mentioned specificity and sensitivity of the various patterns without having calculated them based on ROC curves and AUC analysis”

OUR REPLY: We corrected our use of the terms “specificity” and “sensitivity” and calculated and applied them as appropriate. We also introduced the AUC analysis of the ROC curve to compare the various models. Lines 115-136.

ACADEMIC EDITORS#4: “we recommend that you deposit your laboratory protocols”

OUR REPLY: We have deposited both of them in figshare.com and they have DOI:

1. Okafor O, Ezemba N, Onyishi N, Ezike K. S1_Table.csv. February 2021. doi:10.6084/m9.figshare.14067221.v1 

2. Okafor O, Ezemba N, Onyishi N, Ezike K. S1_R_code. February 2021. doi:10.6084/m9.figshare.14068715.v1

ACADEMIC EDITOR#5: “Please amend your current ethics statement to include the full name of the ethics committee/institutional review board(s) that approved your specific study.”

OUR REPLY: We have done this in the Materials and Methods. Lines 88-93.

ACADEMIC EDITOR#6: “ensure that you have discussed whether all data/samples were fully anonymized before you accessed them”

OUR REPLY: We have included this detail in the Materials and Methods and in the online submission form. Lines 97-100.

ACADEMIC EDITOR#7: “In the ethics statement in the manuscript and in the online submission form, please provide additional information about the patient records/samples used in your retrospective study, including: a) the date range (month and year) during which patients' medical records/samples were accessed.”

OUR REPLY: We have included this information both in the manuscript and in the online submission form. Lines 88-90.

ACADEMIC EDITOR#8: “In your Methods section, please provide additional information about the medical data/samples collected and the demographic details of the human subjects. Please ensure you have provided sufficient details to replicated the analyses such as a table of relevant demographic details.”

OUR REPLY: A new table describing the demographic details of all the human subjects employed in the study has been inserted in the manuscript. Tables 1 and 2 in Results section.

ACADEMIC EDITOR#9: “In your Methods section, please provide a description of the outcomes assessed in your study.”

OUR REPLY: A detailed description of the clinical outcomes of the test patients has been provided in the manuscript. Lines 151-163.

ACADEMIC EDITOR#10: “Please ensure you have discussed any potential limitations of your study in the Discussion, including study design, sample size and/or potential confounders.”

OUR REPLY: We have included this in the Discussion section. Lines 441-445.

ACADEMIC EDITOR#11: “please provide additional information regarding your statistical analyses.”

OUR REPLY: All the statistical methods used were briefly described in the Materials and Methods. Lines 101-138. S1 R_code, our R working script, was deposited online at figshare.com (Available online at: doi:10.6084/m9.figshare.14102849.v1)

ACADEMIC EDITOR#12: “Please include captions for your Supporting Information files at the end of your manuscript, and update any in-text citations to match accordingly.”

OUR REPLY: This has been written out at the end of the manuscript; in-text citations have also been placed. Lines 550-559.

REVIEWER-1#1: “I would suggest to the Authors to report in the Discussion section a brief view of the literature about the discrepancy between the clinical and histological diagnosis reported by different centers”

OUR COMMENT: In the Discussion we have cited 5 earlier studies on this discrepancy and then extensively discussed the outcome and conclusions of one them (Walter AJ et al., Am J Obstet Gynecol. 2001), which indirectly influences our present study. Lines 345-361.

REVIEWER-2#1: “Introduction, line 46, 47: please, cite some possible theory regarding etiology of TE”

OUR COMMENT: We have cited 2 articles on this. Line 44.

REVIEWER-2#2: “Introduction should be shorter”

OUR COMMENT: The Introduction has been shortened from 638 words to 548 words.

REVIEWER-2#3: “Material and methods, from line 106 to 114: this part should be moved in the results part”

OUR COMMENT: These lines relating to the human subjects’ demographics and other related data have been moved to the Results section. Lines 140-164.

REVIEWER-2#4: “Discussion: I would suggest discussing also when TE is impossible to be proved by histology (e.g.: diaphragmatic hernias). For instance, recently it has been reported (Viti A, et al. Endometriosis Involving the Diaphragm: A Patient-Tailored Minimally Invasive Surgical Treatment. World J Surg. 2020 Apr;44(4):1099-1104.) that up to 45% of patient with diaphragmatic endometriosis had no histological diagnosis. Please discuss on this.”

OUR COMMENT: The emphasis of that study was on the surgical treatment of 21 cases of diaphragmatic endometriosis. In their study only 12 cases had samples analysed histologically, of which 10 were classified as endometriosis and 2 as "stromal-only" endometriosis. The other 9 cases were cases of diaphragmatic perforations in the absence of nodules and sample were not taken for histology i.e. not a case of incapacity to histologically detect endometriosis in samples sent. In those 9 cases the diagnosis of endometriosis therefore was based on symptomatology and/or prior history of confirmed abdominal/pelvic endometriosis. Our own study is based entirely on the histological diagnosis of endometriosis; and the surgical approach carried out in our hospital emphasises on thoroughly exploring the thoracic cavity and striving to obtain samples for histological analysis. As a consequence of this thoroughness there was no case of failure to establish a histological diagnosis in all the cases that had diaphragmatic lesions of any form. Thank you for giving me the opportunity to discuss this. We have not included this discussion in the manuscript because, as said earlier, our emphasis here is on making and improving the histological diagnostic yield on samples sent for analysis.

REVIEWER-3#1: “First of all in endometriosis there are endometrioid glands and endometrioid stroma (not endometrial)”

OUR COMMENT: We have made these changes.

REVIEWER-3#2: “periods lines 63-68 are confusing; be concise; lines 68-70”

OUR COMMENT: Thank you for pointing this ambiguity and imprecision out. We have attempted to clearly point out the divergent lines of thought that derive from an initial consideration, thus: “In order to resolve this dilemma, the finding, on one hand, of long duration prior to diagnosis in spite of hospital presentation (possibly to different clinicians) will point to the clinicians as the cause of under-diagnosis. On the other hand, the correct and definitive diagnosis coming only after clinico-pathological reviews necessitated by non-resolution of symptoms, insistence of the clinicians or requests for second opinion will point to the histopathologist as the cause of the underdiagnosis and apparent rarity of the disease. The scenario is further compounded by a finding that less than 50% of laparoscopic biopsies of lesions visually diagnosed as endometriosis were confirmed to be endometriosis on histological study [8].” Lines 58-67.

REVIEWER-3#3: “delete lines 76-83; Insert the following:”

OUR COMMENT: The deletions and insertions have been done. Lines 69-78.

REVIEWER-3#4: “Material and methods: insert from results: The interval between the onset of symptoms 239 and definitive histopathological

diagnosis was 49 weeks (range = 1 week to 300 weeks).”

OUR COMMENT: At the instance of the 2nd Reviewer (REVIEWER-2#3:) we moved this phrase and other demographic aspects of the human subjects to Results section. Lines 144-164.

REVIEWER-3#5: “There are too many figures. Delete figures 2, 4, and 11. Change the order of the rest:”

OUR COMMENT: We deleted the 3 figures and changed the ordering and labels of the rest in accord with this comment. Thank you for the detailed commenting.

REVIEWER-3#6: “Results: Change the order: Nine notable histological features ... Twenty-four cases (92%)... Three histological features ...”

OUR REPLY: We have reordered the paragraphs in accord with this comment. Lines 165-209.

REVIEWER-3#7: Structural modification of 4 paragraphs.

OUR COMMENT: We have implemented all the essential changes of this structural reconstruction into the Discussion.

---

## [Decision Letter · Decision Letter 1]

16 Mar 2021

PONE-D-20-36704R1

Improving the diagnostic recognition of thoracic endometriosis: Spotlight on a new histo-morphological indicator

PLOS ONE

Dear Dr. Okafor,

Thank you for submitting your manuscript to PLOS ONE. After careful consideration, we feel that it has merit but does not fully meet PLOS ONE’s publication criteria as it currently stands. Therefore, we invite you to submit a revised version of the manuscript that addresses the points raised during the review process.

ACADEMIC EDITOR:

-The Materials and Methods section is still insufficent. The lab protocol needs to be better detailed. How were the various histopathological features identified? 

- For the statistical analysis, it is not necessary to explain the tests. The authors have to explain HOW they performed the tests. Results from the paper should be confirmed by other scientists. If others are not able to confirm the results, the study has no value. 

- A R script cannot be presented as supplementary table. Again, data need to be understood by everyone. Please transform the R script in something that is understandable. Moreover, correct S1 R code all over the text. - It is unclear how the cut offs for sensitivity and specificity were established. These explanations should be in the statistical analysis section. 

- A great limit of the paper is that the reliability of the new diagnostic test should be evaluated in a different cohort. Although collecting a new cohort might be difficult, the Authors should declare this limit of the study in the Discussion section.

This novel approach with new histopathologic findings is interesting but the paper still needs substantial ameliorations.

We look forward to receiving your revised manuscript.

Kind regards,

Paola Viganò

Academic Editor

PLOS ONE

Additional Editor Comments (if provided):

The materials and methods section is still insufficent. The lab protocol needs to be better detailed. How were the various histopathological features identified? For the statistical analysis, it is not necessary to explain the tests. The authors have to explain how they performed the tests. Results from the paper should be confirmed by other scientists. If others are not able to confirm the results, the study has no value.

A R script cannot be presented as supplementary table. Again, data need to be understood by everyone. Please transform the R script in something that is understandable. Moreover, correct S1 R code all over the text.

It is unclear how the cut offs for sensitivity and specificity were established. These explanations should be in the statistical analysis section.

A great limit of the paper is that the reliability of the new diagnostic test should be evaluated in a different cohort. Although collecting a new cohort might be difficult, the Authors should declare this limit of the study in the Discussion section

This novel aaproach with new histopathologic findings is interesting but the paper still needs substantial ameliorations

Reviewers' comments:

Reviewer's Responses to Questions

**Comments to the Author**

1. If the authors have adequately addressed your comments raised in a previous round of review and you feel that this manuscript is now acceptable for publication, you may indicate that here to bypass the “Comments to the Author” section, enter your conflict of interest statement in the “Confidential to Editor” section, and submit your "Accept" recommendation.

Reviewer #1: All comments have been addressed

Reviewer #2: All comments have been addressed

2. Is the manuscript technically sound, and do the data support the conclusions?

Reviewer #1: Yes

Reviewer #2: Yes

3. Has the statistical analysis been performed appropriately and rigorously? 

Reviewer #1: N/A

Reviewer #2: Yes

4. Have the authors made all data underlying the findings in their manuscript fully available?

Reviewer #1: Yes

Reviewer #2: Yes

5. Is the manuscript presented in an intelligible fashion and written in standard English?

Reviewer #1: Yes

Reviewer #2: Yes

6. Review Comments to the Author

Reviewer #1: (No Response)

Reviewer #2: The authors significantly improved their paper after revisions and they should be commended.

I appreciate their reply to my last comment and I do agree that their paper is about histological features of thoracic endometriosis, but I think it may be worth to briefly discuss that histological proof of endometriosis is not always possible; as a matter of fact, in some cases diagnosis is made on macroscopic details (e.g. diagphragmatic hernias) and patients’ medical history. As suggested by dr. Triponez and colleagues (Triponez F, Alifano M, Bobbio A et al (2010) Endometriosis related spontaneous diaphragmatic rupture. Interact Cardiovasc Thorac Surg 11(4):485–487) diaphragmatic rupture might be the result of the involution of endometriosis tissue. What should the authors suggest to biopsy in these cases of no evidence of macroscopic endometriosis tissue? How did the authors manage similar cases? Maybe authors should also briefly describe in the results the macroscopical features of endometriosic lesions they resected.

7. PLOS authors have the option to publish the peer review history of their article (what does this mean?). If published, this will include your full peer review and any attached files.

Reviewer #1: **Yes: **Paola Ciriaco

Reviewer #2: **Yes: **Pietro Bertoglio

---

## [Author Response · Author response to Decision Letter 1]

19 Apr 2021

ACADEMIC EDITOR#1: “How were the various histopathological features identified?”

OUR REPLY: We have described our methodology in a detailed manner in Lines 95-101 and referenced the supplementary table (S1 Table) that contains the results.

ACADEMIC EDITOR#2: “For the statistical analysis, it is not necessary to explain the tests. The authors have to explain HOW they performed the tests.”

OUR REPLY: The explanation of the statistical methods have been removed. In Lines 102-119 of Material and methods we fully described how the statistical analysis was performed and referenced the supplementary table (S2 Table) that contains the results.

ACADEMIC EDITOR#3: “A R script cannot be presented as supplementary table. Again, data need to be understood by everyone. Please transform the R script in something that is understandable. Moreover, correct S1 R code all over the text.”

OUR REPLY: We removed the R script as supplementary data and erased all reference to it in the article text.

ACADEMIC EDITORS#4: “It is unclear how the cut offs for sensitivity and specificity were established. These explanations should be in the statistical analysis section.”

OUR REPLY: We omitted the use of R code to obtain the best cutoff points but did it manually from the supplementary tables provided; we then described in detail how this was done in Lines 257-262 of the Results section.

ACADEMIC EDITOR#5: “A great limit of the paper is that the reliability of the new diagnostic test should be evaluated in a different cohort. Although collecting a new cohort might be difficult, the Authors should declare this limit of the study in the Discussion section.”

OUR REPLY: We have included this limitation in Lines 430-431 of Discussion section.

REVIEWER-2: “diaphragmatic rupture might be the result of the involution of endometriosis tissue. What should the authors suggest to biopsy in these cases of no evidence of macroscopic endometriosis tissue? How did the authors manage similar cases? Maybe authors should also briefly describe in the results the macroscopical features of endometriosic lesions they resected.”

OUR REPLY: We have briefly described the macroscopic features in Lines 149-151 of the Results section and have suggested what to biopsy in cases of no evidence of macroscopic endometriosis tissue in Lines 323-331 of Discussion section.

---

## [Editor Report · Decision Letter 2]

26 Apr 2021

Improving the diagnostic recognition of thoracic endometriosis: Spotlight on a new histo-morphological indicator

PONE-D-20-36704R2

Dear Dr. Okafor,

We’re pleased to inform you that your manuscript has been judged scientifically suitable for publication and will be formally accepted for publication once it meets all outstanding technical requirements.

Kind regards,

Paola Viganò

Academic Editor

PLOS ONE

Additional Editor Comments (optional):

The Authors have addressed the issues raised
---

## [Editor Report · Acceptance letter]

3 May 2021

PONE-D-20-36704R2 

Improving the diagnostic recognition of thoracic endometriosis: Spotlight on a new histo-morphological indicator. 

Dear Dr. Okafor:

I'm pleased to inform you that your manuscript has been deemed suitable for publication in PLOS ONE. Congratulations! Your manuscript is now with our production department. 

Kind regards, 

on behalf of

Dr. Paola Viganò 

Academic Editor

PLOS ONE